# Cinnamic Acid Attenuates Peripheral and Hypothalamic Inflammation in High-Fat Diet-Induced Obese Mice

**DOI:** 10.3390/pharmaceutics14081675

**Published:** 2022-08-11

**Authors:** Aden Geonhee Lee, Sora Kang, Suyeol Im, Youngmi Kim Pak

**Affiliations:** 1Phillips Exeter Academy, Exeter, NH 03833, USA; 2Department of Neuroscience, Graduate School, Kyung Hee University, Seoul 02447, Korea; 3Department of Physiology, College of Medicine, Kyung Hee University, Seoul 02447, Korea

**Keywords:** cinnamic acid, inflammation, high-fat diet, monocytes, hypothalamus

## Abstract

Obesity is closely linked to chronic inflammation in peripheral organs and the hypothalamus. Chronic consumption of a high-fat diet (HFD) induces the differentiation of Ly6c^high^ monocytes into macrophages in adipose tissue, the liver, and the brain, as well as the secretion of pro-inflammatory cytokines. Although cinnamon improves obesity and related diseases, it is unclear which components of cinnamon can affect macrophages and inflammatory cytokines. We performed in silico analyses using ADME, drug-likeness, and molecular docking simulations to predict the active compounds of cinnamon. Among the 82 active compounds of cinnamon, cinnamic acid (CA) showed the highest score of ADME, blood–brain barrier permeability, drug-likeness, and cytokine binding. We then investigated whether CA modulates obesity-induced metabolic profiles and macrophage-related inflammatory responses in HFD-fed mice. While HFD feeding induced obesity, CA ameliorated obesity and related symptoms, such as epididymal fat gain, insulin resistance, glucose intolerance, and dyslipidemia, without hepatic and renal toxicity. CA also improved HFD-induced tumor necrosis factor-α, fat deposition, and macrophage infiltration in the liver and adipose tissue. CA decreased Ly6c^high^ monocytes, adipose tissue M1 macrophages, and hypothalamic microglial activation. These results suggest that CA attenuates the peripheral and hypothalamic inflammatory monocytes/macrophage system and treats obesity-related metabolic disorders.

## 1. Introduction

Obesity rates have been rapidly increasing worldwide, becoming a significant health and financial burden. Obesity is a medical disorder characterized by excessive fat accumulation. It is associated with severe health concerns, such as type 2 diabetes mellitus, dyslipidemia, cardiovascular disease, sleep apnea, and Alzheimer’s disease, shortening life expectancy, and increasing morbidity [1,2].

It has been generally accepted that obesity is a condition of chronic low-grade systemic inflammation that is a major contributor to the development of obesity-associated insulin resistance and metabolic derangement [3]. Obesity induces hypoxic damage in adipose tissue, which secretes chemokines, providing a chemotactic gradient to recruit monocytes. Monocytes, especially Ly6c^high^ inflammatory monocytes, infiltrate adipose tissue and the liver and change to inflammatory macrophages [4,5]. Newly recruited macrophages activate and release pro-inflammatory cytokines, such as tumor necrosis factor (TNF)-α, interleukin (IL)-6, interferon (IFN)-γ, as well as chemokines, consequently leading to elevated levels of inflammatory markers in blood and organs [3,6]. The activated adipose tissue macrophages have been recognized as the primary source of systemic low-grade inflammation and insulin resistance [7], and activated Kupffer cells also increase hepatic insulin resistance and steatosis [8].

Likewise, consumption of a high-fat diet (HFD) induces low-grade inflammation in the hypothalamus, which is characterized by the expansion of the monocyte-derived macrophage pool in the hypothalamic arcuate nucleus (ARC) [9]. Microglia, CNS macrophages, have an origin distinct from peripheral macrophages and are important innate immune cells in the CNS that sense pathogenic invasion or tissue damage. The activated microglia disrupt the blood–brain barrier (BBB), which enables the inflammatory cytokines to cross the BBB and results in neuroinflammation and, consequently, neurodegenerative diseases [10]. Therefore, modulating the macrophages and inflammation in adipose tissue, the liver, and the brain could be an effective therapeutic strategy for obesity-associated complications [11].

The bark of the Cinnamomum genus (Cinnamon) of aromatic trees has been used since ancient times as a flavor additive, and also as an antimicrobial [12], anti-parasitic [13], anti-inflammatory [14], antioxidant [15], antidiabetic, anti-hyperlipidemic [16,17,18], and anticancer agent [19,20]. Clinical and experimental studies have reported that Cinnamon improves obesity and related diseases, such as prediabetes, diabetes, hypertension [21], dyslipidemia, cognitive function [22], and also protects dopaminergic neuronal function in Parkinson’s disease mouse model [23].

Cinnamon consists of various physicochemical compounds, including cinnamaldehyde, cinnamate, Camphor, and cinnamic acid (CA), as well as numerous essential oils, such as trans-cinnamaldehyde, cinnamyl acetate, eugenol, α-caryophyllene, β-caryophyllene, α-terpineol, α-terpineol, and α-thujene [24,25]. However, it has not been studied which components of Cinnamon can affect macrophage and inflammatory cytokine and which cellular and molecular mechanisms are involved. CA, a primary compound of Cinnamon, improves obesity, diabetes [26], dyslipidemia, and also crosses the BBB fairly well allowing it to protect the brain [27]. This study investigated the effects of CA on obesity-induced metabolic changes and macrophage-related inflammatory responses, focusing on the brain microglia in high-fat diet (HFD)-fed C57BL/6 mice.

## 2. Materials and Methods

### 2.1. Computational Biology Experiment

#### 2.1.1. Data Preparation

A total of 82 active components in the *Cinnamomum zeylanicum* and *Cinnamon cassia* were manually extracted from PubMed (http://www.ncbi.nlm.nih.gov/pubmed, accessed on 12 March 2021), Science Direct (http://www.sciencedirect.com/, accessed on 12 March 2021), Scopus (http://www.scopus.com/, accessed on 12 March 2021), Scirus (http://www.scirus.com/, accessed on 12 March 2021), and Google Scholar (http://www.scholar.google.com/, accessed on 12 March 2021), and previous reports (https://doi.org/10.1080/10942912.2017.1369102, accessed on 12 March 2021), (https://doi.org/10.1155/2014/642942, accessed on 12 March 2021) (Appendix A).

#### 2.1.2. Active Component Prediction

After the active component screening, an absorption, distribution, metabolism, and excretion (ADME) and drug-likeness evaluation system was applied to select potential active components. This prediction was made using ADMElab (https://admet.scbdd.com/home/index/, accessed on 13 March 2021), and SwissADME (http://www.swissadme.ch/, accessed on 13 March 2021). We set three pharmacokinetic parameters to identify the active and safe components of the Cinnamon; gastrointestinal (GI) absorption, blood–brain barrier (BBB) permeability, and cytochrome 450 (CYP450). This study used GI absorption ≥ −5 and BBB permeability ≥ 0.9 as the screening criteria for active components, and CYP450s (CYP3A4, CYP2D6, CYP2C9, CYP1A2, and CYP2C19) for safe compounds. Lipinski’s oral drug-likeness [28] was predicted with five rules: (1) molecular weight ≤500 Dalton, (2) hydrogen bond donors ≤ 5, (3) hydrogen bond acceptors ≤ 10, and (4) an octanol-water partition coefficient log *p* ≤ 5. The highest bioavailability score (>85%) was also applied for predicting the permeability and bioavailability properties of Cinnamon compounds before conducting in vivo experiments. Active components that met these criteria were regarded as candidate components for subsequent molecular docking analysis.

#### 2.1.3. Molecular Docking

To investigate the interaction of Cinnamon bioactive compounds with cytokines secreted by macrophages, we analyzed the binding affinity and the compounds’ active-binding site. We selected 15 major compounds of Cinnamon from the results of ADME, BBB permeability, and drug-likeness as ligands, and macrophages as binding receptors to evaluate the compounds’ binding capacity to macrophages. We analyzed TNF-α for assessing their impact on the inflammatory cytokines. The 3D structural data of compounds including CA (CID 444539), ferulic acid (CDI 445858), *p*-coumaric acid (CID 637542), *p*-hydroxybenzoic acid (CID 135), and salicylic acid (CID 338) were obtained from PubChem (https://pubchem.ncbi.nlm.nih.gov/, accessed on 22 March 2021). The 3D crystal structures of migration inhibitory factor (MIF, PDBID: 1GD0) [29] secreted by proinflammatory M1 macrophages, matrix metalloproteinase-12 (MMP-12, PDBID: 1JIZ) [30] secreted by anti-inflammatory M2 macrophages, and TNF-α (PDBID: 2AZ5) [31] were acquired from the PDB database (https://www.rcsb.org, accessed on 22 March 2021). Molecular docking was simulated using the AutoDock open-source program. PyMOL was used to estimate binding site and affinity, and to visualize binding structure [32]. The AutoDock Vina was used to calculate binding scores (kcal/mol) [33]. Lower binding energies indicated better docking [34]. According to the molecular docking analysis results, we focused on the effect of CA on macrophage and inflammation.

### 2.2. Animal Experiment

#### 2.2.1. Chemicals and Reagents

We purchased CA (≥99% purity, CAS#: 140-10-3) and bovine serum albumin (BSA) from Sigma-Aldrich (St. Louis, MO, USA), and fetal bovine serum (FBS), and 0.25% trypsin-ethylenediaminetetraacetic acid (EDTA) from Gibco (Carlsbad, CA, USA). Type 2 collagenase was purchased from Worthington (Columbus, OH, USA), and deoxyribonuclease I was from Roche (Basel, Switzerland).

#### 2.2.2. Experimental Design

Six-week-old male C57BL/6 mice (*n* = 5~6/group) were kept in a moisture-controlled condition with a 12 light/12 dark cycle and provided water and diet. After one week of acclimation, mice were fed ad libitum with either standard normal chow (NC, Research Diets D12450B, 10% kcal fat) or a high-fat diet (HFD, Research Diets D12492, 60% kcal fat) for six weeks. After six weeks of NC or HFD feeding, we divided the mice into four groups according to diet; NC group (*n* = 6), HFD group (*n* = 6), HFD plus CA 100 mg/kg group (*n* = 5), HFD plus CA 200 mg/kg group (*n* = 5). CA groups were orally administered once a day for eight weeks, while the NC and HFD groups received saline. Dosages of CA in treatment regimen were decided based on toxicity test and other studies. The median lethal doses (LD50) values of CA were >5000 mg/kg in mice (Galleria Chemica) [35]. The toxicity test of CA using four doses (50, 100, 500, and 1000 mg/kg body weight) demonstrated no mortality and no changes in clinical signs such as external appearance, behavior, and motility [35]. Another obesity-related study used 30–100 mg/kg of CA [36,37]. Therefore, we decided high doses (100 mg/kg and 200 mg/kg) considering BBB permeability of CA sufficient to inhibit hypothalamic inflammation.

At week 14, we sacrificed the mice and measured the weights of the epididymal fat pad and liver. Animal maintenance and treatment were carried out in accordance with the Principles of Laboratory Animal Care (NIH publication No. 85-23, revised 1985) and the Animal Care and Use guidelines of Kyung Hee University, Seoul, Korea. The Animal Research Ethics Committee of Kyung Hee University, Seoul, Korea, approved this study (KHSASP-20-163).

#### 2.2.3. Metabolic Phenotype Measurements

The body weight of each mouse was measured at the beginning and the end of the experiment. Food intake was calculated by total food consumption in every cage during the day. Then the calorie intake was calculated by multiplying the gram of consumed food by calories per gram of the diet type. Metabolic measurements were performed at 11 weeks for oral glucose tolerance test (OGTT), at 12 weeks for oral fat tolerance test (OFTT), and at 13 weeks for homeostatic model assessment for insulin resistance (HOMA-IR). For OGTT, the mice were fasted overnight for 14 h, weighed, and tested for fasting blood glucose using a blood glucose glucometer (ACCU-CHEK, Roche, Basel, Switzerland), then orally administered with glucose (2 g/kg body weight). Blood glucose from the lateral tail was measured at 30, 60, 120, and 180 min after oral glucose administration. For OFTT, mice were fasted overnight and triglyceride (TG) concentrations in plasma were measured using the Accutrend Plus (Roche, San Francisco, CA, USA). The mice were orally administered with olive oil (Sigma, St. Louis, MO, USA; 4 mL/kg body weight), then TG was measured at 0, 120, 240, and 360 min after oil loading. For HOMA-IR measurement, mice were fasted overnight, weighed, and then tested for fasting blood glucose using a glucose meter (ACCU-CHEK Performa, Seoul, Korea) and fasting insulin by enzyme-linked immunosorbent assay (Crystal Chem, Elk Grove Village, IL, USA,). Finally, HOMA-IR was calculated according to the formula: HOMA-IR = (fasting glucose (mg/dL) × fasting blood insulin (ng/mL))/22.5.

#### 2.2.4. Biochemical Assays

At week 14, blood was collected from hearts under diethyl ether anesthesia before the sacrifice. Clinical parameters including total cholesterol, low-density lipoprotein cholesterol (LDL-C), high-density lipoprotein cholesterol (HDL-C), phospholipid, free fatty acid (FFA), aspartate aminotransaminase (AST), alanine aminotransaminase (ALT), and creatinine levels were determined as described [38].

#### 2.2.5. Serum TNF-α and IL-6 Protein Levels

The serum concentration of inflammatory cytokines, TNF-α and IL-6, were measured using mouse TNF-α and mouse IL-6 ELISA kit (BMS607-3 and BMS603-2, Invitrogen, Carlsbad, CA, USA), according to the manufacturer’s protocol.

#### 2.2.6. Preparation of Stromal Vascular Cells (SVCs)

Stromal vascular cells (SVCs) were isolated from epididymal fat pads using a well-established collagenase-based method [39]. Mouse epididymal fat pads were removed, weighed, and minced in phosphate-buffered saline (PBS, Gibco, Waltham, MA, USA) containing 2% bovine serum albumin (BSA, Gibco, USA). Collagenase type 2 (10 mg/mL, Worthington) and deoxyribonuclease I (2 mg/mL, Roche, Indianapolis, IN, USA) were added to the samples before incubation at 37 °C for 20 min with shaking. Digested adipose tissue was filtered through a 100 μm cell strainer (BD Biosciences, San Diego, CA, USA) to remove non-digested adipose tissue and then added 2% BSA/PBS and 5 mM EDTA. The suspension was centrifuged at 300 g for 5 min, and the pellet was resuspended in 2% FBS (Sigma-Aldrich)/PBS. The suspension was then filtered through a 100 μm nylon mesh to remove unnecessary tissue and centrifuged at 300 g for 5 min. The pellets containing SVCs were resuspended with 2% FBS/PBS, and prepared for staining.

#### 2.2.7. Cell Staining and Fluorescence Activated Cell Sorting (FACS)

Isolated SVCs and blood cells were incubated with 1% Fc Block (BD Biosciences) and then stained with fluorophore-conjugated antibodies in the dark for 20 min. The following fluorophore-conjugated antibodies (BioLegend, San Diego, CA, USA) were used for staining: CD11b- phycoerythrin Cyanine7, F4/80-APC, CD11c-phycoerythrin, CD206-FITC, CD45-APC Cyanine7, CD3-FITC, CD4-PerCP Cyanine5.5, and Ly6c-APC. To identify adipose tissue macrophages (ATMs), we analyzed the CD11b^+^ and F4/80^+^ ATMs, CD11c^+^ M1 ATMs, CD206^+^ M2 ATMs after CD45^+^, CD3^−^, CD19^−^, NK1.1^−^, and TER119^−^ gating; for lymphocytes, CD3^+^ T cells, CD4^+^ T cells, CD8^+^ T cells after CD45^+^ gating; for blood Ly6c monocytes, Ly6c^+^ and CD11b^+^ cells after CD45^+^ gating. After staining, cells were resuspended in FACS buffer and counted using BD Canto flow cytometer (BD Biosciences). The number of the immune cells was analyzed by the FlowJo FACS analysis program (Tree Star, Inc., Ashland, OR, USA).

#### 2.2.8. Histological Analysis of the Liver and Epididymal Fat Pad

Liver and epididymal fat pads were fixed in 4% formaldehyde (PFA) solution and created paraffin blocks. We cut each block into 5 μm thick slices using a microtome and placed each on a slide. For hematoxylin and eosin (H&E) and immunohistochemistry (IHC) staining, the slides of two tissues per animal were deparaffinized with xylene and gradient rehydrated. The images were captured with a computer-connected BX50 microscope (Olympus Optical, Tokyo, Japan). Finally, we measured the adipocyte size of the epididymal fat pad and the lipid droplet area in the liver with ImageJ.

#### 2.2.9. Brain Tissue Preparation and IHC Staining

Whole-brain tissues were dissected from the skull, postfixed overnight with 4% paraformaldehyde in 0.1 M phosphate buffer (PB) at 4 °C, and then stored in 30% sucrose solution in 0.05 M PBS at 4 °C until they sank. The brains were frozen-sectioned on a Cryostat (Microsystems AG, Leica, Wetzlar, Germany) with 30 µm thick coronal sections and stored in a cryoprotectant (25% ethylene glycol, 25% glycerol, 0.2 M PB, and water) at 4 °C until use [40]. All sections were collected in six separate series and processed for immunostaining as previously described [41].

Brain coronal sections (30 μm in thickness) containing hypothalamus [42] were incubated with rabbit anti-glial fibrillary acidic protein (anti-GFAP, 1:5000; Neuromics, Edina, MN, USA) or rabbit anti-ionized calcium binding adaptor molecule 1 (anti-Iba-1, 1:1000; Wako, Osaka, Japan), followed by staining with biotinylated anti-rabbit IgG and an avidin-biotin peroxidase complex (ABC) standard kit (Vector Laboratories, Burlingame, CA, USA). Signals were detected by incubating sections with 0.5 mg/mL 3,3′-diaminobenzidine (Sigma, St. Louis, MO, USA) in 0.1 M PBS containing 0.003% H_2_O_2_. To quantify GFAP- or Iba-1-positive cells, stained brain sections were imaged under a bright-field microscope (Olympus Optical, Tokyo, Japan).

#### 2.2.10. Quantitation of Activated Microglia and Astrocytes

To quantify resting and activated microglia and astrocytes in the hypothalamus, coronal brain sections (30 μm thickness) were collected (five sections/series) and labeled with anti-Iba-1 or anti-GFAP antibodies. For quantitative analysis, the sections labeled with Iba-1- or GFAP were digitized and manually counted within preselected fields (500 × 400 μm) of hypothalamus (two fields/animal). Activated microglia and astrocytes were classified and counted according to their morphologies. Briefly, resting microglia and astrocytes displayed small compact somata that exhibited long, thin, and ramified processes. In contrast, activated microglia and astrocytes exhibited marked cellular hypertrophy and retraction of processes, such that the length of the process was less than the diameter of the soma compartment. Cells were sampled only if the nucleus was visible within the sectional plane and if the cell profiles exhibited distinct delineated borders [43].

#### 2.2.11. Quantitative Reverse Transcription-PCR (qRT-PCR)

We acquired epididymal fat pads and livers and stored them at –70 °C. RNA of epididymal fat pads was extracted using a Mini RNA Isolation IITM (Zymo Research, Orange, CA, USA). Total RNA from liver tissue was isolated using Trizol reagent (Invitrogen, Carlsbad, CA, USA). Total RNA (2 μg) was reverse transcribed using MMLV reverse transcriptase (Promega, Madison, WI, USA) and 10 pM oligo dT primers (Invitrogen), according to each manufacturer’s instructions.

The levels of mRNA were determined by real-time qRT-PCR using cDNA, SYBR PCR master mix, and primers on Roter-Gene Q (Qiagen, Hilden, Germany) with 2× AmpiGene^®^ qPCR green Mix Lo-ROX (Enzo Biochem, New York, NY, USA) at 95 °C for 10 min, followed by 45 cycles of 95 °C for 5 s, 60 °C for 15 s, and 72 °C for 20 s. Measurements were performed in duplicate for each sample. The quantity of mRNA was normalized by simultaneous measurement of nuclear DNA encoding glyceraldehyde-3-phosphate dehydrogenase (GAPDH). The relative gene expression was determined using the 2^−ΔΔCt^ method [44]. Relative levels of mRNA expression are presented as fold changes compared to those under the control condition, the expression in the normal chow (NC)-fed control group as 1. Sequences of primers include TNF-α (5′-AAGCCTGTAGCCCACGTCGTA-3′, and 5′-GGCACCACTAGTTGGTTGTCTTTG-3′), F4/80 (5′-CTTTGGCTATGGGCTTCCAGTC-3′ and 5′-GCAAGGAGGACAGAGTTTATCGTG-3′), IL-6 (5′-AACGATGATGCACTTGCAGA-3′; 5′-GAGCATTGGAAATTGGGGTA-3′), glyceraldehyde-3-phosphate dehydrogenase (GAPDH; 5′-ACTCCACTCACGGCAAATTC-3′, and 5′-TCTCCATGGTGGTGAAGACA-3′).

#### 2.2.12. Statistical Analysis

Data shown are expressed as mean ± standard error of the mean (SEM). Statistical significance was evaluated by one-way analysis of variance (ANOVA) with Tukey post-hoc testing analysis using InStat (GraphPad Software, San Diego, CA, USA). A two-tailed *p*-value less than 0.05 was considered statistically significant.

## 3. Results

### 3.1. ADME Predictions

ADME properties of the Cinnamon compounds were determined based on GI absorption, BBB permeability, CYP, and Lipinski’s oral drug-likeness. Among a total of 82 active components in Cinnamon (List in Appendix A), 35 compounds were predicted to be well-absorbed in the GI, passively cross the BBB, and met Lipinski’s oral drug-likeness. The drug–drug interaction predicted by inhibition of CYP resulted in 15 compounds. Only five compounds, including CA, ferulic acid, p-coumaric acid, p-hydroxybenzoic acid, and salicylic acid, showed the highest bioavailability score (>85%) (Appendix A). Physicochemical properties, lipophilicity, water solubility, and pharmacokinetics of the selected five Cinnamon compounds are summarized in Appendix A. Among five compounds, ferulic acid showed highest GI absorption, and CA showed highest BBB permeability, while *p*-coumaric acid, *p*-hydroxybenzoic acid, and salicylic acid showed low BBB permeability (Appendix A). We selected CA with its high score of BBB permeability, as a candidate with anti-inflammatory activity against HFD-induced peripheral and hypothalamic inflammation and macrophage activation.

### 3.2. Molecular Docking

Before the initiation of the in vivo experiment, molecular docking in silico analysis was performed to analyze whether CA binds to TNF-α and other cytokines secreted from macrophages [31]. The AutoDock program predicted that CA has the highest binding affinity with TNF-α (−5.9 kcal/mol), MIF (−7.2 kcal/mol), and MMP-12, a macrophage-secreted elastase (−6.6 kcal/mol) (Figure 1). Molecular docking binding affinities of five cinnamon components to TNF-α, MIF, and MMP-12 are presented in Appendix A.

### 3.3. Metabolic Phenotypes

Through in silico screening, CA was found to act on macrophage activation and modulate inflammation in HFD-induced obesity. To confirm its anti-inflammatory effects on HFD-induced obesity, mice were fed with HFD for 14 weeks with and without CA administration, as summarized in Figure 2A. The HFD control group had remarkably increased body weight (BW) compared to the normal chow (NC) group (50.82 ± 1.31 g vs. 27.84 ± 0.69 g, *p* < 0.001) (Figure 2B). Administration with CA 100 mg/kg (CA100) and 200 mg/kg (CA200) significantly decreased the average changes of body weight (ΔBW) by 14% and 24%, respectively, as compared to the HFD-fed control group (*p* < 0.01) (Figure 2C).

The oral calorie intake of HFD-fed mice was significantly increased compared to the NC group (*p* < 0.001), but there was no difference between the HFD control and CA groups (Figure 2D). This indicates the body weight reduction by CA treatment was not due to calorie intake changes. HFD also increased epididymal white adipose tissue (EWAT) and liver weights (*p* < 0.001 vs. NC), but CA treatment reduced their weights significantly compared to HFD control (*p* < 0.05 and *p* < 0.01, Figure 2E,F).

### 3.4. Glucose Metabolism and Insulin Resistance

Fasting blood glucose (FBG) was increased in the HFD control group (226.1 ± 17.2 mg/dL), and decreased by CA (CA100: 177.6 ± 8.8 mg/dL, *p* < 0.05, and CA200: 154.0 ± 14.2 mg/dL, *p* < 0.01) (Figure 2G). Similarly, fasting insulin was increased by HFD and decreased by CA (Figure 2H). CA treatment significantly improved HOMA-IR, an index to determine insulin resistance, compared to HFD control (*p*  < 0.01) (Figure 2I). Oral glucose tolerance test (OGTT) confirmed the effects of CA treatment on glucose intolerance (Figure 2J). After glucose load, CA treatment significantly reduced blood glucose level and area under the curve (AUC) in HFD groups (*p* < 0.05) (Figure 2J,K).

**Figure 2 pharmaceutics-14-01675-f002:**
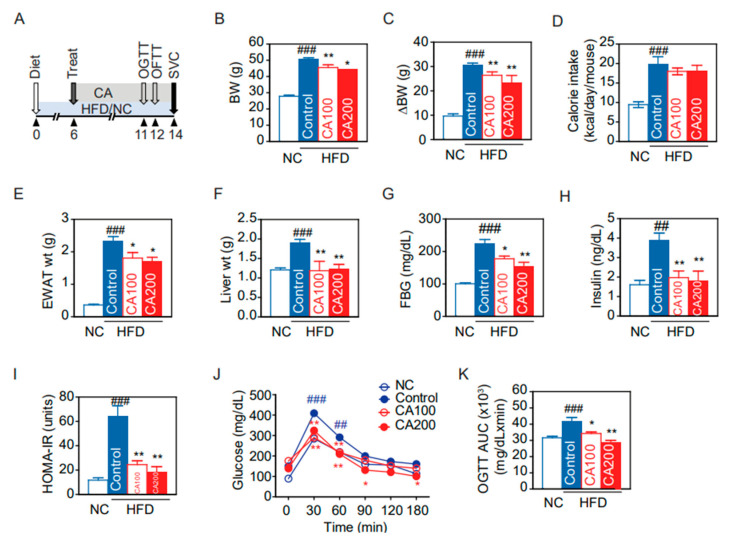
Changes of glucose and metabolic profiles by CA in HFD mice: (**A**) experimental scheme, (**B**) body weight (BW) at the end of the experiment, (**C**) changes of BW, (**D**) oral calories intake, (**E**) epididymal white adipose tissue (EWAT) weight, (**F**) liver weight, (**G**) fasting blood glucose (FBG), (**H**) fasting insulin, (**I**) homeostatic model assessment of insulin resistance (HOMA-IR), (**J**) time-course changes of blood glucose in oral glucose tolerance test (OGTT), and (**K**) area under the curve (AUC) of OGTT. Data are expressed as the mean ± SEM (*n* = 5). ## *p* < 0.01, ### *p* < 0.001 vs. NC, and * *p* < 0.05, and ** *p* < 0.01 vs. HFD-control. NC, normal chow; Control, HFD-control; CA100, HFD plus CA 100 mg/kg; CA200, HFD plus CA 200 mg/kg.

### 3.5. Lipid Metabolism

Since insulin resistance is closely related with lipid metabolism, we analyzed the lipid profile and performed an oral fat tolerance test (OFTT) to evaluate the postprandial lipid metabolism alterations. HFD increased total cholesterol (total-C) and LDL-cholesterol (LDL-C) (209.1 ± 9.5 mg/dL and 36.0 ± 2.1 mg/dL, respectively. *p* < 0.001) and decreased HDL-cholesterol (HDL-C) (108.6 ± 8.9 mg/dL, *p* < 0.001) (Figure 3A–C). CA200 decreased the LDL-C (28.0 ± 4.0 mg/dL, *p* < 0.05) but not total-C. Interestingly, CA100 and CA200 significantly increased HDL-C (*p* < 0.001) (Figure 3C). After oil load for OFTT, blood TG increased at 0, 120 and 240 min in the HFD control group (150.0 ± 8.5, 447.7 ± 53.0 and 307.0 ± 20.3 mg/dL, respectively) (Figure 3D). CA100 decreased TG at 240 min (*p* < 0.05) and CA200 decreased at 120, 240, 360 min (*p* < 0.001 or *p* < 0.05) (Figure 3D). CA treatment also decreased OFTT AUC of TG significantly (*p* < 0.01) compared to HFD control (Figure 3E). However, the effects of CA treatments on phospholipid (PL) and free fatty acid (FFA) were negligible (Figure 3F,G).

### 3.6. Toxicity Biochemical Parameters

To assess if CA induces hepatic and renal toxicity, serum hepatic or renal injury markers were analyzed. HFD elevated AST and ALT levels of the liver enzyme significantly, and CA treatment restored the HFD-induced increase in serum ALT (*p* < 0.01, Figure 3H). The serum creatinine did not change in all groups (Figure 3I), suggesting that CA administration did not induce adverse toxic effects on liver and kidney.

### 3.7. Expression of Inflammatory Cytokines

TNF-α and IL-6 are major pro-inflammatory cytokines that are predominantly produced by macrophages. We measured the serum TNF-α and IL-6 using ELISA and then analyzed mRNA levels in EWAT and the liver. HFD increased serum TNF-α levels by 3.2-fold (*p* < 0.001), and CA decreased the HFD-induced TNF-α by approximately 50% (*p* < 0.05) (Figure 3J). HFD also increased serum IL-6 levels by 2.1-fold (*p* < 0.05), but CA administration did not significantly reduce serum IL-6 levels (Figure 3K). Real-time qRT-PCR demonstrated that HFD up-regulated the mRNA levels of TNF-α by 9.5-fold and by 2.2-fold, respectively, in EWAT and the liver (Figure 3L,M). CA treatment significantly reduced TNF-α mRNA in EWAT, but not in the liver (*p* < 0.05). HFD enhanced IL-6 mRNA levels by 5.3-fold in EWAT (*p* < 0.001), but CA administration did not significantly reduce IL-6 mRNA levels (Figure 3L). In the liver, IL-6 mRNA levels were unchanged in all groups (Figure 3M).

**Figure 3 pharmaceutics-14-01675-f003:**
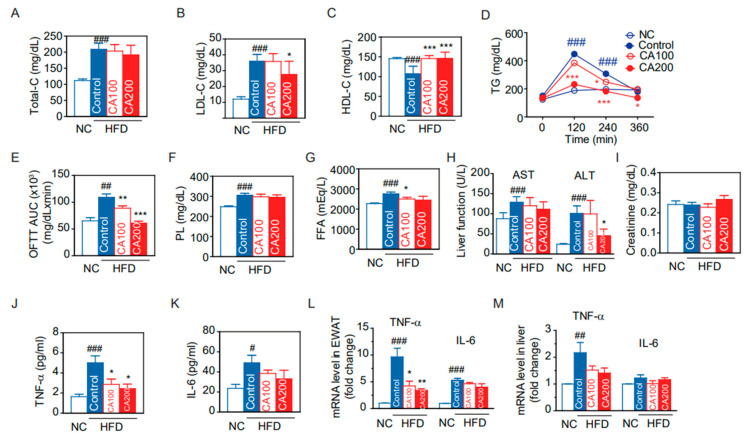
Changes of lipid and metabolic profiles by CA in HFD mice: (**A**) total-cholesterol (Total-C), (**B**) LDL-cholesterol (LDL-C), (**C**) HDL-cholesterol (HDL-C), (**D**) time course of serum TG in oral fat tolerance test (OFTT), (**E**) area under the curve (AUC) of OFTT, (**F**) phospholipids, (**G**) free fatty acid (FFA), (**H**) liver function marker enzyme activities of AST and ALT, (**I**) creatinine, and (**J**,**K**) ELISA. Serum protein levels of TNF-α (**J**), and IL-6 (**K**). (**L**,**M**) Real-time qRT-PCR. TNF-α and IL-6 mRNA levels in EWAS (L) or liver (M). Data are expressed as the mean ± SEM (*n* = 5). # *p* < 0.05, ## *p* < 0.01, ### *p* < 0.001 vs. NC, and * *p* < 0.05, ** *p* < 0.01, and *** *p* < 0.001 vs. HFD-control. NC, normal chow; Control, HFD-control; CA100, HFD plus CA 100 mg/kg; CA200, HFD plus CA 200 mg/kg.

### 3.8. Inflammatory Monocytes

Under inflammatory conditions, Ly6c^high^ monocytes differentiate into M1 pro-inflammatory macrophages in the local cytokine environment [45]. We analyzed the differences in peripheral blood CD11b^+^ Ly6c^high^ cells among NC and HFD groups. Proportions of CD11b^+^ Ly6c^high^ and CD11b^+^ Ly6c^low^ cells were determined as shown in the density plot (Figure 4A). The CD11b^+^ Ly6c^high^ population in the HFD-control group was markedly increased while decreased by CA treatment (*p* < 0.001, Figure 4B). However, CD11b^+^ Ly6c^low^ monocytes, neutrophils, CD4^+^ T cells, CD8^+^ T cells, and CD4/CD8 ratio were not altered in all groups (Figure 4, Appendix A).

### 3.9. Fat Deposition and Macrophage Infiltration in Liver and Adipose Tissue

Fat deposition in the liver and adipose tissue results in insulin resistance, leading to increased levels of circulating inflammatory cytokines. The extent of fat accumulation in the liver and EWAT of HFD mice was analyzed by measuring the area of hepatic lipid droplet (LD) and the size of adipocytes (Figure 5). HFD increased the area of hepatic LD (*p* < 0.001, Figure 5A,B) and adipocyte size (hypertrophied adipocyte) (*p* < 0.001, Figure 5D,E), whereas CA treatment reduced them to the level of the NC group. These results correlate with liver weights in study groups.

Because the recruitment of hepatic macrophages was increased in HFD mice, we examined macrophage infiltration in CA-treated liver. Hepatic F4/80-positive cells (Figure 5A) and mRNA levels of F4/80 (Figure 5C) were increased in HFD-control compared to NC (*p* < 0.05) and were decreased by CA treatments (statistically not significant, *p* = 1.3). On the other hand, adipose tissue macrophages (ATMs) are the predominant leukocytes in fat and contribute to obesity-induced inflammation [46]. Immunochemical staining of EWAT for F4/80 indicated a greater presence of macrophages in HFD-control than in NC (Figure 5D), and CA treatment restored F4/80-positive cells to the levels of NC (*p* < 0.05). This result was confirmed by quantification of mRNA levels of F4/80 in adipose tissue (Figure 5F).

### 3.10. Adipose Tissue Macrophages (ATMs)

Flow cytometry and flow-assisted cell sorting techniques permit the immunophenotyping, quantification, and purification of ATM populations from multiple adipose tissue depots. Proper isolation, quantification, and characterization of ATM phenotypes are critical for understanding their role in obesity-induced metabolic diseases [46]. The percentage of total ATMs significantly increased in the HFD-control group than in the NC group (*p* < 0.001, Figure 6A). The percentage of ATMs in the CA-treated groups was significantly lower than in the HFD-control group (*p* < 0.05, Figure 6A). Similarly, HFD increased the proportion of CD11c^+^ ATMs, M1 pro-inflammatory macrophages, that differentiate from CD11b^+^ Ly6c^+^ monocytes and express TNF-α and IL-6 (*p* < 0.001, Figure 6B). CA treatments lowered the percentage of CD11c^+^ M1 ATMs (*p* < 0.01, Figure 6B). The percentage of CD206^+^ ATMs, resident M2 anti-inflammatory macrophages, was significantly more decreased in the HFD-control than in the NC group (*p* < 0.001, Figure 6C). The CA treatments significantly increased the percentage of CD206^+^ M2 ATMs compared to the HFD-control (*p* < 0.05, Figure 6C).

### 3.11. Inflammation in the Hypothalamus

HFD causes inflammation in peripheral organs, which is, in turn, associated with inflammation in brain areas, such as the hippocampus and the hypothalamus. In particular, inflammation in the arcuate nucleus (ARC) in the mediobasal hypothalamus happens rapidly, before obesity is established [42]. Hypothalamic inflammation causes changes in ARC neurons and is responsible for diet-induced obesity and insulin resistance. To check whether CA treatment modulates hypothalamic inflammation, HFD feeding-induced activation of microglia and astrocytes was determined by immunohistochemical staining of coronal sections of brain.

Having found that CA treatment inhibited peripheral inflammation in HFD-fed mice, we focused on hypothalamic inflammation. HFD caused microglia accumulation in the ARC (Figure 7A,B), with the microglia, labeled with Iba-1, displaying the typical morphology of activated cells with enlarged soma size (Figure 7A). CA treatment during HFD-feeding period decreased the number of Iba1-positive cells (Figure 7A,B) and blunted the characteristic morphological changes of microglial activation. When astrocyte accumulation was determined in the ARC, with GFAP-labeled astrocytes, results similar to those of Iba-1 positive cells were obtained (Figure 7). The number or morphology of microglia and astrocytes was not altered in the hippocampus, suggesting that microglia activation was not generalized to the whole brain (data not shown).

## 4. Discussion

This is the first study to find that cinnamic acid (CA) modulates inflammatory monocytes/macrophages and inflammatory cytokines in adipose tissue, the liver, and the hypothalamus.

Cinnamon, the inner bark of trees from the genus Cinnamomum, has two main varieties: *Cinnamomum zeylanicum* and *Cinnamon cassia* (also known as Chinese Cinnamon). Cinnamon mainly contains various vital oils and resinous compounds, such as CA, cinnamaldehyde, cinnamate, and numerous essential oils. Cinnamon has been used as a common spice worldwide and is considered a remedy for respiratory, digestive, and gynecological disorders. Recently, several randomized clinical trials have addressed the effects of Cinnamon lowering fasting plasma glucose, glucose tolerance, glycated hemoglobin (HbA1c), and the rate of progression from prediabetes to T2D [47,48,49]. In addition, a meta-analysis reported that Cinnamon lowers total cholesterol and TG without any significant effects on LDL-C and HDL-C [50]. However, which bioactive compounds of Cinnamon are pharmacologically and biologically effective have not yet been clarified. Therefore, we conducted an in silico approach to predict the active compounds of Cinnamon according to ADME and drug-likeness, both of which can profile drug candidates, helping minimize the costs involving experimental, preclinical, and clinical studies.

ADME predictions showed that 15 compounds, including cinnamaldehyde, cinnamyl acetate, camphor, borneol, ferulic acid, vanillin, p-coumaric acid, p-hydroxybenzoic acid, salicylic acid, and CA passed the GI absorption, BBB permeability and oral drug-likeness test with no drug–drug interactions. GI absorption can estimate the concentration of ingested compounds that could be absorbed from the intestine and enter the bloodstream [51]. BBB permeability can predict which Cinnamon compounds are able to pass the BBB and affect inflammation in the brain. Lipinski’s oral drug-likeness can predict the pharmacological or biological properties of Cinnamon compounds [52]. CYP450 can predict drug-drug interactions because CYP450 enzymes play an essential role in metabolizing drugs, while the alteration of CYP450 could change the metabolism of other drugs by interacting with them, resulting in side effects. Although there are more than 50 CYP450 enzymes, the CYP1A2, CYP2C9, CYP2C19, CYP2D6, and CYP3A4 enzymes metabolize 90 percent of drugs [53]. Therefore, we analyzed the CYP test with 5 CYP450 enzymes, even though drug interaction cannot be perfectly predicted. For example, *p*-coumaric acid passed the CYP test, but it inhibits platelet activity in vitro and in vivo [54]. Thus, if co-administered with anti-coagulation drugs, *p*-coumaric acid could induce life-threatening side effects. The bioavailability score is calculated with a combination of the total charge, topological polar surface area (TPSA), and Lipinski’s rule, then classified into four categories: 11%, 17%, 56%, or 85% [55]. This process enables fast screening and the selection of the best chemicals and molecules before medicinal chemistry research. In this study, five compounds with high bioavailability (>85%), which include CA, *p*-ferulic acid, coumaric acid, *p*-hydroxybenzoic acid, and salicylic acid, were selected among the 82 compounds of Cinnamon. These results suggest that the five selected compounds could be major pharmacologically and biologically effective safe properties of Cinnamon, which may be helpful to future pharmacological study.

The accumulation of macrophages in various tissues of obese mice and humans and the increase in macrophage-derived pro-inflammatory cytokines, such as TNF-α in the circulating blood, are the starting point of the chronic low-grade inflammation that contributes to metabolic diseases [2,56,57]. Thus, we simulated the molecular docking of the five selected active compounds of Cinnamon with MIF, MMP-12, and TNF-α (Figure 1). Among five candidate compounds, CA showed the highest binding affinity with MIF, MMP-12, and TNF-α. Given the in silico analysis with ADME and binding affinity to pro-inflammatory cytokines, the CA compound is likely to be a putative inhibitor of macrophage activation. To evaluate these molecular docking results, experimental validations are required. As expected, our HFD feeding animal study showed that CA markedly decreased the macrophages in adipose tissue, the liver, and the brain hypothalamus. Furthermore, CA decreased TNF-α in serum, adipose tissue, and the liver. The present study expands our understanding concerning the main bioactive compound and the underlying mechanisms of the anti-hyperglycemic and anti-hyperlipidemic effects of Cinnamon.

CA has been reported to have various biological effects such as antibacterial, antitumor, antimicrobial, and antioxidant, in addition to improving obesity, diabetes, hyperlipidemia, glucose intolerance, insulin secretion, and memory deficiency [26,58,59]. This study found that CA suppressed weight gain for all six weeks compared to the HFD-fed control group without calorie intake reduction (Figure 2). CA also reduced epididymal fat and liver weight gain, adipocyte size and hepatic fat accumulation. CA improves insulin resistance, glucose intolerance, and lipid metabolism (Figure 3). As expected from the in silico results, CA did not cause any toxicity to hepatic and renal function in the in vivo experiments.

Adiposity and adipocyte hypertrophy caused by overnutrition induces hypoxia, increases oxidative stress by reactive oxygen species production, local inflammatory cytokines and chemokine production, and induces the recruitment of monocytes, which develop into tissue macrophages. In lean condition, Ly6c^low^ monocytes differentiate into M2 type macrophages, which normally reside in all metabolic tissues including adipose tissue, the liver, and the brain [60,61,62] for wound repair, tissue remodeling, maintaining homeostasis, and produce anti-inflammatory mediators such as IL-10 and TGF-β. But in an obese state, Ly6c^high^ monocytes infiltrate tissues and differentiate into M1 type macrophages, which produce inflammatory cytokines, such as TNF-α, IL-6, IL-1β, and IL-18, and drive insulin resistance [63,64,65]. In this study, CA decreased the number of inflammatory Ly6c^high^ monocytes (Figure 4), macrophages (Figure 5 and Figure 6), and pro-inflammatory TNF-α in the blood, adipose tissue, and the liver (Figure 3). These effects of CA on monocytes, macrophages, and inflammatory cytokines may be the main underlying mechanism to improve insulin resistance, glucose intolerance, and lipid metabolism.

In particular, we discovered that CA treatment decreases the number of activated microglia (Iba-1 positive) and astrocytes (GFAP-positive) in the hypothalamus (Figure 7). HFD-induced hypothalamic inflammation occurs specifically in the ARC, where microglia initiate an inflammatory response by releasing proinflammatory cytokines and chemokines [66]. Under physiological conditions, microglia typically function to clear cellular and neuronal debris with a pro-and anti-inflammatory balance. On the other hand, obesity destroys synapses and then reprograms microglia as a pro-inflammatory response to eliminate synaptic debris from the hypothalamus, resulting in hypothalamic microglial activation, leading to cognitive decline as seen in the HFD-fed mice [2,67,68,69]. Astrocytes, another glial cell type that is abundant in the CNS, supports neuronal function. Astrocytes control synaptic function and plasticity, and their foot processes cover blood vessels and help maintain the BBB integrity. Upon prolonged HFD consumption, the number of activated astrocytes is increased to sustain hypothalamic inflammation. This process is referred to as reactive astrogliosis. Although astrocyte–neuron interactions have been shown to protect against neurotoxicity, reactive astrocytes release proinflammatory mediators that induce neuronal injury. Cross-talk between microglia and astrocytes may play an important role in HFD-induced hypothalamic inflammation. Activated microglia may urge astrocytes to change their phenotype from the resting (non-reactive) to the reactive type by secreting pro-inflammatory molecules, such as IL-1α and TNFα. Conversely, activated astrocytes may regulate microglial activity and contribute to sustained microglial activation. Therefore, during obesity-associated inflammation, the hypothalamus may be one of the most sensitive organs, and ARC neurons become victims of hypothalamic inflammation. Dysfunctional ARC neurons have an impaired ability to sense peripheral metabolic signals, leading to obesity through increased food intake and decreased energy expenditure.

This current study, combined with the ADME results, suggests that BBB-permeable CA may have a direct beneficial effect on the inflammation of the hypothalamic ARC. Then, CA may control energy balance and appetite through the regulation of neuropeptides for food intake in the ARC. CA-mediated TNF-α inhibition in circulating blood may also indirectly modulate microglial activation in the hypothalamus. It has long been recognized that a chronic low-grade inflammation and an activation of the immune system are involved in the pathogenesis of obesity-related insulin resistance and type 2 diabetes [70]. Obesity-induced inflammation suppresses the insulin-signaling pathway, making the human body less responsive to insulin and increasing the risk of insulin resistance. CA inhibited TNF-α levels in blood, which in turn can reduce macrophage levels in adipose tissue, the liver, and the hypothalamus. CA reduced hypothalamic inflammation, which can increase appetite, weight gain and fat accumulation. These combined effects of CA may lead to improvements in insulin resistance, glucose intolerance, and lipid metabolism. Anti-inflammatory agents such as salicylates, TNF-α antagonists, IL-1β antagonists, leukotrienes, and CA may be useful in developing therapeutics for obesity-related diseases, including insulin resistance, type 2 diabetes, and atherosclerotic cardiovascular disease [71]. We believe that CA-mediated inhibition of both peripheral and hypothalamic inflammation sheds light on the development of novel therapeutics to treat obesity-induced metabolic diseases.

## 5. Conclusions

The current study provides new evidence that CA, a primary compound of Cinnamon, modulates inflammation in peripheral organs as well as in the brain. The results suggest that CA could be applied to treat obesity and related metabolic disorders, especially regulating energy homeostasis. Further studies will be necessary to confirm that CA may be an excellent candidate to control obesity-associated inflammatory diseases.

## Figures and Tables

**Figure 1 pharmaceutics-14-01675-f001:**
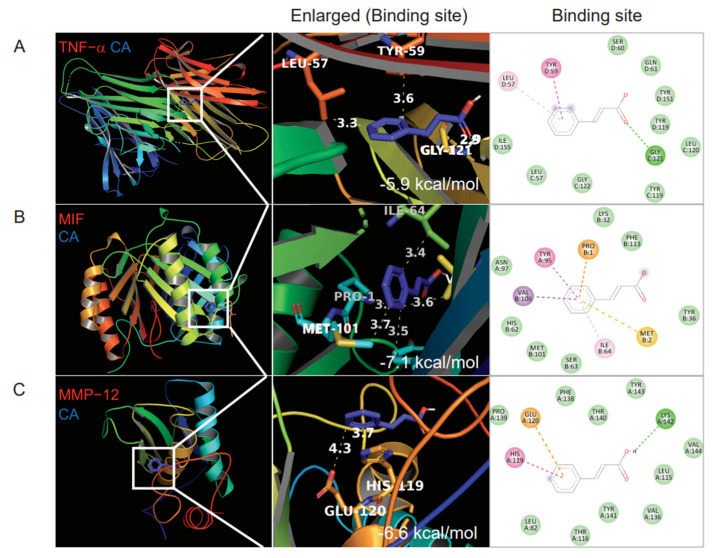
Molecular docking binding affinity of cinnamic acid (CA) for TNF−α, migration inhibitory factor (MIF), and elastase MMP−12. (**A**) CA binding to TNF−α, (**B**) CA binding to MIF, (**C**) CA 12 binding to MMP. Three-dimensional structures of TNF-α (PDBID: 2AZ5), MIF (PDBID: 1GD0), and MMP-12 (PDBID: 1JIZ) were obtained from the PDB database. CA binding sites in the left panel are enlarged as indicated by white boxes. The binding score is displayed in the lower right corner of the second panel.

**Figure 4 pharmaceutics-14-01675-f004:**
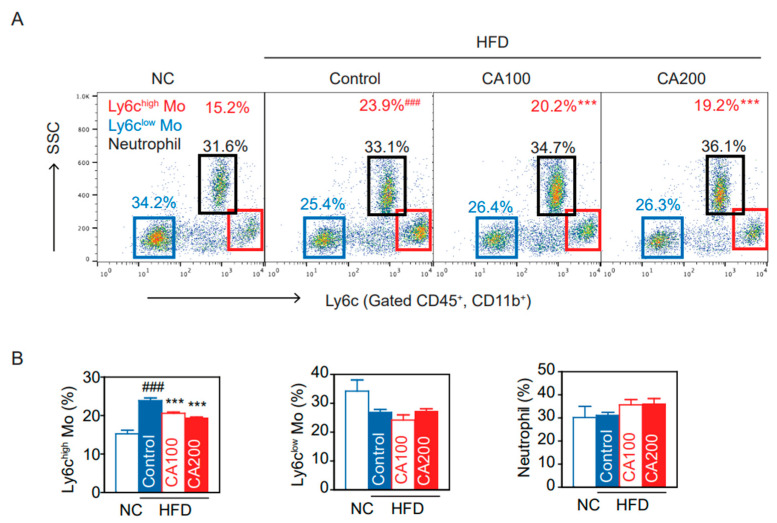
CA restored blood monocyte (Mo) subtypes in HFD mice: (**A**) flow cytometry of blood monocytes (Ly6c Mo) and neutrophils, and (**B**) percentage of Ly6c^high^ monocytes, Ly6c^low^ monocytes, and neutrophils. Data are expressed as the mean ± SEM (*n* = 5). ### *p* < 0.001 vs. NC and *** *p* < 0.001 vs. HFD-control. NC, normal chow; Control, HFD-control; CA100, HFD plus CA 100 mg/kg; CA200, HFD plus CA 200 mg/kg.

**Figure 5 pharmaceutics-14-01675-f005:**
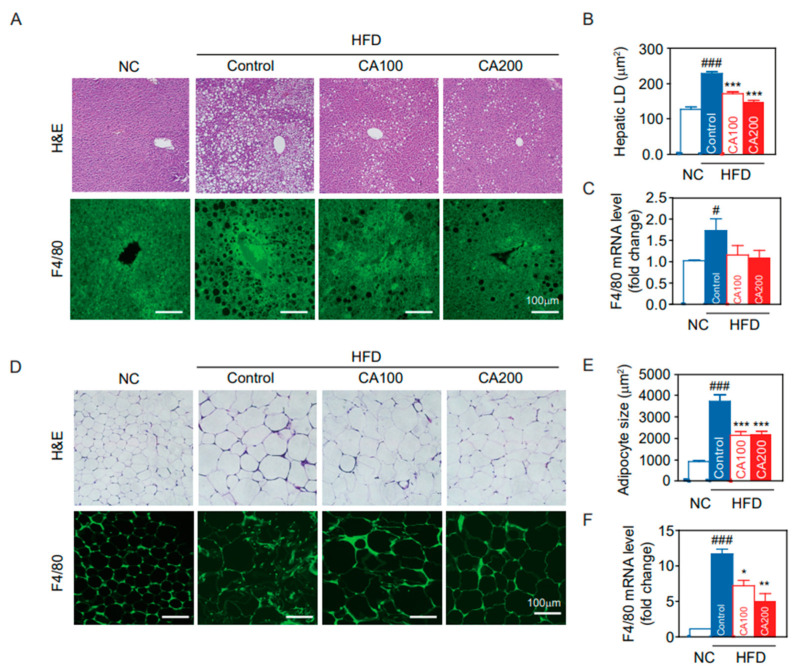
Histological analysis of the liver and epididymal fat pad (EWAT). (**A**) liver H&E staining and F4/80 immunostaining, (**B**) hepatic lipid droplet (LD) area, (**C**) F4/80 mRNA levels in the liver, (**D**) EWAT H&E staining and F4/80 immunostaining, (**E**) adipocyte size of the EWAT, and (**F**) F4/80 mRNA levels in the EWAT. Data are expressed as the mean ± SEM (*n* = 5). # *p* < 0.05, ### *p* < 0.001 vs. NC, and * *p* < 0.05, ** *p* < 0.01, and *** *p* < 0.001 vs. HFD-control. NC, normal chow; Control, HFD-control; CA100, HFD plus CA 100 mg/kg; CA200, HFD plus CA 200 mg/kg.

**Figure 6 pharmaceutics-14-01675-f006:**
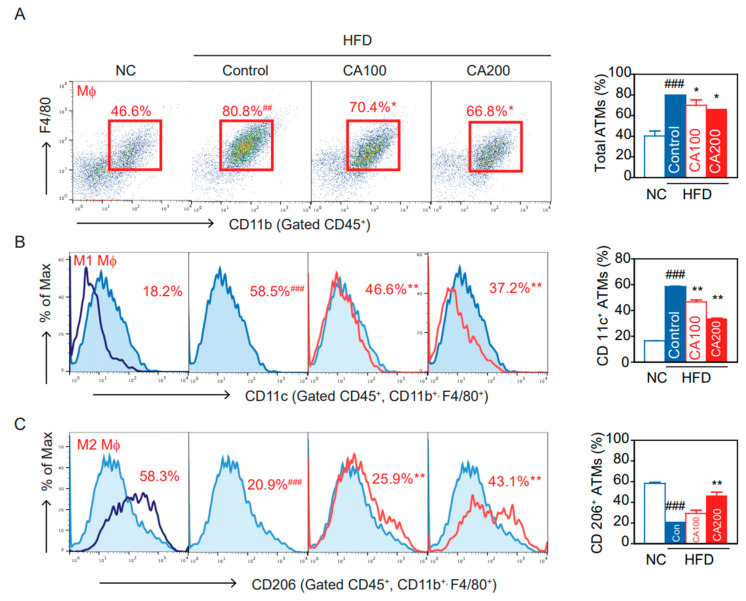
Effects of CA on macrophages and subtypes of adipose tissue macrophages (ATM) in HFD mice: (**A**) flow cytometry of CD45+, CD11b+, and F4/80+ ATM, and percentage of total ATM, (**B**) percentage of CD11c+ inflammatory ATM, and (**C**) percentage of CD206^+^ anti-inflammatory ATM. Data are expressed as the mean ± SEM (*n* = 5). ## *p* < 0.01, ### *p* < 0.001 vs. NC, and * *p* < 0.05, ** *p* < 0.01 vs. HFD-control. NC, normal chow; Control, HFD-control; CA100, HFD plus CA 100 mg/kg; CA200, HFD plus CA 200 mg/kg.

**Figure 7 pharmaceutics-14-01675-f007:**
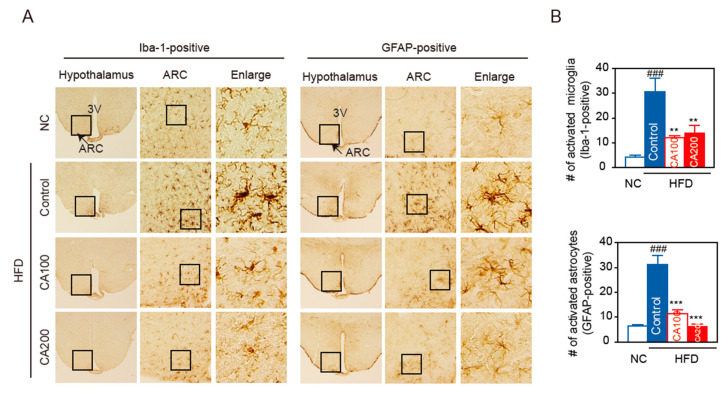
CA attenuates HFD-induced hypothalamic inflammation. (**A**) Representative images and higher magnification inset of hypothalamus of HFD-fed mice. Hypothalamic sections were immuno-stained with Iba-1 or GFAP antibodies. The arcuate nucleus (ARC) area (arrow, inset box) and the third ventricle (3V) are indicated. Box area in ARC was enlarged to view the cellular morphology. (**B**) Quantification of activated microglia (Iba-1 positive cells) and activated astrocytes (GFAP-positive cells) in the ARC. Data are expressed as the mean ± SEM (*n* = 6). ### *p* < 0.001 vs. NC, and ** *p* < 0.01, and *** *p* < 0.001 vs. HFD-control. NC, normal chow; Control, HFD-control; CA100, HFD plus CA 100 mg/kg; CA200, HFD plus CA 200 mg/kg.

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
