# Peer review of "Cinnamic Acid Attenuates Peripheral and Hypothalamic Inflammation in High-Fat Diet-Induced Obese Mice"

_pharmaceutics, 2022, doi:10.3390/pharmaceutics14081675_

Round 1

Reviewer 1 Report

In this study, Lee et al examined the effects of cinnamic acid (CA) in obesity-induced metabolic changes and inflammatory mediators using a high fat diet-fed mouse model. The authors found that CA ameliorated HFD-induced epididymal fat gain, insulin resistance, glucose tolerance and dyslipidemia without adverse effects. Macrophage infiltration in liver and adipose tissue as well as HFD-induced inflammatory mediators such as TNF-alpha in blood, adipose tissue and the liver, were also decreased when mice were treated with CA, suggesting CA as a potential therapeutic for obesity-related metabolic disorders. The manuscript is well written, though it requires minor grammatical error checks. Their observations are interesting and reports out significant findings, but lack complete mechanistic understanding.  I have following comments that are recommended to be addressed before considering for publication.

1)    It is not clear how does CA suppress macrophage and monocyte infiltration in HFD-induced mice? The authors also need to discuss how reduced inflammatory response leads to improve insulin resistance, glucose intolerance and lipid metabolism.

2)    What are the signaling cascades that CA utilizes to exert anti-obesity or anti-inflammatory effects?

3)    It is unclear how did authors decide dosages of CA in their treatment regimen.

4)    How is CA relevant in clinical setting for the treatment of female patients suffering from such metabolic disorders?

5)    HFD-fed mice is known to develop severe nephropathy. It would be interesting to see histological analysis of kidney sections demonstrating the degrees of glomerulosclerosis and tubular injury in CA-treated vs untreated mice.

Reviewer 2 Report

Accept in current form 

Author Response

Thank you.

Reviewer 3 Report

This is a well designed and solid study. The authors investigated the active compounds in Cinnamon that have beneficial effect on high fat diet induced obesity. They used both computation and animal models to show that Cinnamic acid ameliorates HFD induced obesity and inflammation. The results are clearly presented and easy to understand. The data support their conclusions.  

Author Response

Thank you for your nice comment.

Round 2

Reviewer 1 Report

Overall, the authors have improved the manuscript and presented and discussed the results in clearer manner.